# Article

# Effect of Filler Morphology on the Electrical and Thermal Conductivity of PP/Carbon-Based Nanocomposites

Marta Zaccone [1,*], Alberto Frache [2], Luigi Torre [3], Ilaria Armentano [4] and Marco Monti [1]

1   Proplast, Via Roberto di Ferro 86, 15122 Alessandria, Italy; marco.monti@proplast.it
2   Department of Applied Science and Technology, Polytechnic of Turin, INSTM Research Unit,
    Viale Teresa Michel 5, 15121 Alessandria, Italy; alberto.frache@polito.it
3   Civil and Environmental Engineering Department, University of Perugia, UdR INSTM, Strada di Pentima 4,
    05100 Terni, Italy; luigi.torre@unipg.it
4   Department of Economics, Engineering, Society and Business Organization (DEIM), University of Tuscia,
    Largo dell'Università snc, 01100 Viterbo, Italy; ilaria.armentano@unitus.it
*   Correspondence: marta.zaccone@proplast.it; Tel.: +39-01311859755

**Abstract:** In this paper, we studied the effect of different carbon-based nanostructures on the electrical and mechanical properties of polypropylene (PP) nanocomposites. Multi-walled carbon nanotubes (MWCNT), expanded graphite (EG), and two different carbon black nanoparticles (CB) have been dispersed at several weight contents in the polymer matrix through a melt extrusion process. The produced nanocomposites have been used to obtain samples for the characterization by injection molding. The dispersion of the nanoparticles in the matrix has been evaluated by scanning electron microscopy (SEM) analysis. The electrical characterization has been performed both in DC and in AC configuration. The mechanical properties have been evaluated with both tensile test and impact strength (Izod). The thermal conductivity has been also evaluated. As a result, MWCNTs are the nanoadditive with the lowest electrical percolation threshold. This allows MWCNT nanocomposite to drastically change the electrical behavior without a significant embrittlement observed with the other nanoadditives. However, CB with the lowest surface area allows the highest conductivity, even though at a high particle content. EG has a limited effect on electrical properties, but it is the only one with a significant effect on thermal conductivity.

**Keywords:** carbon nanostructures; polymer nanocomposites; dielectric properties; electrical properties; thermal conductivity



## 1. Introduction

In recent years, the interest in electrically conductive polymer nanocomposites has significantly increased, due to the importance of such polymers in the digitalization process, which is affecting our society, especially for what concerns the manufacturing of smart electronic devices [1]. Carbon-based nanoadditives represent one of the most important families to change the electrical behavior of the polymers in which they are embedded [2–5]. They can be exploited for a variety of industrial applications such as electrostatic discharge [6,7], sensors, and electromagnetic interference (EMI) shielding devices [8–12].

Different nanoadditives belong to this family and they can be diversified according to the number of their dimensions (1D, 2D or 3D) in the range of nano-scale. Carbon black nanoparticles (CB), with all dimensions in the nano-scale, are the most used additive to make polymers conductive [3,13,14]. CB has a relatively low cost and an adequate conductive efficiency. However, based on the purity level and particle size [15], CB nanoparticles are available with different properties and cost. Approximately 90% of CB is used in rubber applications, 9% as a pigment in printing inks or coatings, and the remaining 1% as an ingredient in plastic components [16,17]. Expanded graphite (EG) is a partially exfoliated graphite, characterized by a lamellar structure, and it represents a compromise between

graphite and graphene in terms of electrical and thermal properties and cost [4,18,19]. Carbon nanotubes (CNTs), due to their chain-like structure, are able to form a conductive pathway within the polymer matrix, which is typically more effective if compared with the ones produced by other additives [5,20–22].

The percolation threshold and the electrical conductivity of the nanocomposites are affected by several factors, like the additive volume fraction and its distribution and orientation in the polymeric matrix. Nonetheless, the carbon-based nanostructure features, such as size (aspect ratio and/or surface area), shape, and morphology, have a central role in the electrical and mechanical behavior of the nanocomposites [23–27]. For instance, Gulrez et al. [24] demonstrate how the use of diverse carbon nanostructures as additives for polyolefin polymers and the adoption of different processing methods to fabricate conductive nanocomposites can affect the electrical properties and percolation thresholds of the final components. Moreover, in his review, Huang et al. [28] analyzes the effect of CB not only on the electrical behavior of polymeric nanocomposites, but also on their mechanical properties, which can be negatively affected when this nanoparticle is used at very high level.

Beside the evaluation of DC electrical properties, several studies have focused their attention to dielectric spectroscopy, i.e., the study of frequency-dependent AC electrical properties. Their evaluation can be exploited as a valuable support for understanding the correlation between the morphology of a polymer nanocomposite and its electrical behavior, which may be masked with the DC approach [29–35]. As an example, Monti et al. [35] have carried out an in-depth study of the relationship among crystalline structure, morphology and processing conditions of PP-based nanocomposites adopting the dielectric spectroscopy technique as measurement method.

In this paper, we report a comparative study on the effect of different carbon nanostructures on the electrical, thermal conductivity, and mechanical properties of the PP nanocomposites in which they are embedded. We have been able to perform an in-depth comparison of the nanostructure efficiency, in particular taking advantage of the dielectric spectroscopy approach. As a result, a clear overview of the effect of carbon nanostructures in PP nanocomposites has been obtained.

Although a similar approach has been proposed by other authors [2,36], the present study is particularly focused on industrially available and cost-effective solutions which can inspire and drive innovation even in the industrial community, with a higher potential impact.

## 2. Materials and Methods

An injection molding grade PP (Moplen RP348R), produced by LyondellBasell, was selected as polymer matrix. It is a random copolymer with melt flow index of 25 g/10 min (230 °C—2.16 kg) and density of 0.9 g/cm$^3$. Multi-walled carbon nanotubes (Nanocyl™ NC7000, Nanocyl, Sambreville, Belgium, MWCNT hereinafter), two carbon black (Ensaco® 250G and Ensaco® 350G, Imerys, Paris, France, hereinafter referred to as CB-65 and CB-770, respectively), and expanded graphite (Timrex C-Therm™ 001, Imerys, Paris, France EG hereinafter) were selected as carbon-based additives.

Several formulations were prepared, homogeneously mixing additives and polymer in a co-rotating twin-screw extruder, Leistritz 27E (Leistritz, Nürnberg, Germany). The screws have a diameter (D) of 27 mm and a length of 40D. The screw speed was maintained constant at 220 rpm and the temperature profile was set in the range of 190–200 °C. The obtained nanocomposites were injection molded, using an Engel VC 500/120 press (Engel, Schwertberg, Austria), with a screw diameter D of 40 mm. The temperature of the mold was set at 25 °C and the flow rate was 70 cm$^3$/s. Rectangular-shaped samples (100 × 140 × 2 mm$^3$ in size) were produced. Specimens for the Izod impact test (80 × 10 × 4 mm$^3$ in size) have been also produced. In Table 1, specific physical features of the used carbon-based nanoadditives and added contents in the formulations are reported.

**Table 1.** Carbon-based nanoadditives: main properties and contents.

| Carbon-Based Additives | Main Properties and Contents |
|---|---|
| Multi-walled carbon nanotubes (MWCNT) Nanocyl™ NC7000 Nanocyl (Belgium) | Average diameter: 9.5 nm Average length: 1.5 μm Metal oxide: 10% Surface area (BET method): 250–300 m$^2$/g Contents: 1–2–3–4–5–6–7 wt% |
| Carbon Black (CB-65) Ensaco® 250G Imerys (France) | Surface area (BET method): 65 m$^2$/g Ash content: 0.01% Pour density: 170 kg/m$^3$ Contents: 7.5–10–12.5–15–17.5–20–25 wt% |
| Carbon Black (CB-770) Ensaco® 350G Imerys (France) | Surface area (BET method): 770 m$^2$/g Ash content: 0.03% Pour density: 135 kg/m$^3$ Contents: 5–7.5–10–12.5 wt% |
| Expanded Graphite (EG) Timrex C-Therm™ 001 Imerys (France) | Ash content: <0.3% Scott (bulk) density: 0.15 (g/cm$^3$) Contents: 5–10–15 wt% |

Carbon-based additives were thermally characterized through thermogravimetric analysis (TGA) using a Q500 TA Instruments (TA Instruments, New Castle, DE, USA). Tests were performed in a temperature range from 50 °C to 900 °C in an oxidative atmosphere (air), with a heating rate of 10 °C/min (gas flow of 60 mL/min).

DC electrical properties were evaluated performing bulk resistivity measurements in the through-thickness direction, according to ASTM D257 standard and using a Keithley 6517B electrometer with a Keithley 8009 test fixture (Tektronix Inc, Beaverton, OR, USA). Each test was performed on three different specimens.

AC dielectric characterization was performed in the frequency range $2 \times 10^1 \div 1 \times 10^6$ Hz, in order to evaluate the frequency dependent AC electrical conductivity ($\sigma_{AC}$). The test was performed using a HP 4284A precision LCR meter (Hewlett-Packard, Palo Alto, CA, USA). AC electrical conductivity is calculated taking into account the geometrical parameters of each single specimen according to the equation $\sigma_{AC}(\omega) = \frac{1}{|Z(\omega)|}\frac{t}{A}$, where $|Z(\omega)|$ is the module of the impedance, $t$ is the thickness of the specimen, and $A$ is the area of the electrode used for the measurements (in this case $A = 490$ mm$^2$).

AC dielectric test was performed in the frequency range $1 \times 10^6 \div 1 \times 10^9$ Hz, for the evaluation of the real part of permittivity ($\varepsilon'$). The test was performed using a HP 4291A RF impedance analyzer (Hewlett-Packard, Palo Alto, CA, USA), with a 16,453 A dielectric test fixture.

Tensile test was performed according to UNI EN ISO 527-2 standard. The used dynamometer was a Zwick Roell Z010 model (Zwick Roell, Ulm, Germany). The load cell has a maximum capacity of 10 kN. A speed test of 1 mm/min was applied for the evaluation of the elastic modulus and a speed test of 5 mm/min was applied for the evaluation of all the other tensile parameters. The results are reported in terms of elastic modulus, stress at yield and break, elongation at yield and break. Test specimens (type 5A) were cut from rectangular-shape injection-molded samples.

Izod notched impact test was performed according to UNI EN ISO 180/A, using a pendulum ATS FAAR Impacts-15 (ATS FAAR, Segrate (MI), Italy). The applied impact energy, selected considering the mechanical properties of the nanocomposites, was 1 J and the impact speed of the hammer was 3.46 m/s. The A-type notch has been obtained according to ISO 2818 using a 6816 Notchvis Instron-CEAST equipment (Instron, Norwood, MA, USA). The results are reported using the impact strength value, defined as impact energy absorbed in breaking a sample, referred to the original cross-sectional area.

Thermal conductivity tests were performed in the through-thickness direction using an ISOMET 2104 (Applied Precision Ltd., Rača, Slovakia) equipment. The test is based

on transient probe technique. A round surface probe with a diameter of 65 mm has been used, in which heat pulse is generated for a time interval and the temperature response is analyzed by means of a temperature sensor connected to the heater. Each test was performed on three different specimens.

The morphology of the samples has been investigated by means of scanning electron microscopy (SEM), by using a Zeiss LEO-1450VP by Zeiss (beam voltage: 20 kV; working distance: 15 mm) (Carl Zeiss, Oberkochen, Germany). The specimens were cryo-fractured in liquid nitrogen and coated with a thin layer (<10 nm) of gold before observation, using a Sputter Coater—Emitech K550 (Quorumtech, Laughton, East Sussex, UK).

## 3. Results and Discussion

Figure 1 reports the results of the thermogravimetric analysis in oxidative atmosphere of the different carbon-based nanoadditives. As it is observable, all the tested carbon nanostructures show high thermal resistance with the peak of the derivative weight loss (DTG) higher than 500 °C. More in detail, this peak increases from MWCNTs (611 °C) to CB-65 (681 °C), to CB-770 (744 °C) and to EG (788 °C). Moreover, it can be observed that all the carbon nanostructures display a single step of thermal degradation. This behavior indicates a high purity and a strong thermal stability of all the studied carbon nanostructures in general and in particular, at the typical manufacturing temperature for PP nanocomposites, which are in the range of 190–220 °C. No relevant residue content is observable at 900 °C for both CBs and EG. On the contrary, MWCNT shows a residue of around 10 wt% at 900 °C. It corresponds to the percentage of metal oxide present as a catalysis, as declared by the producer.

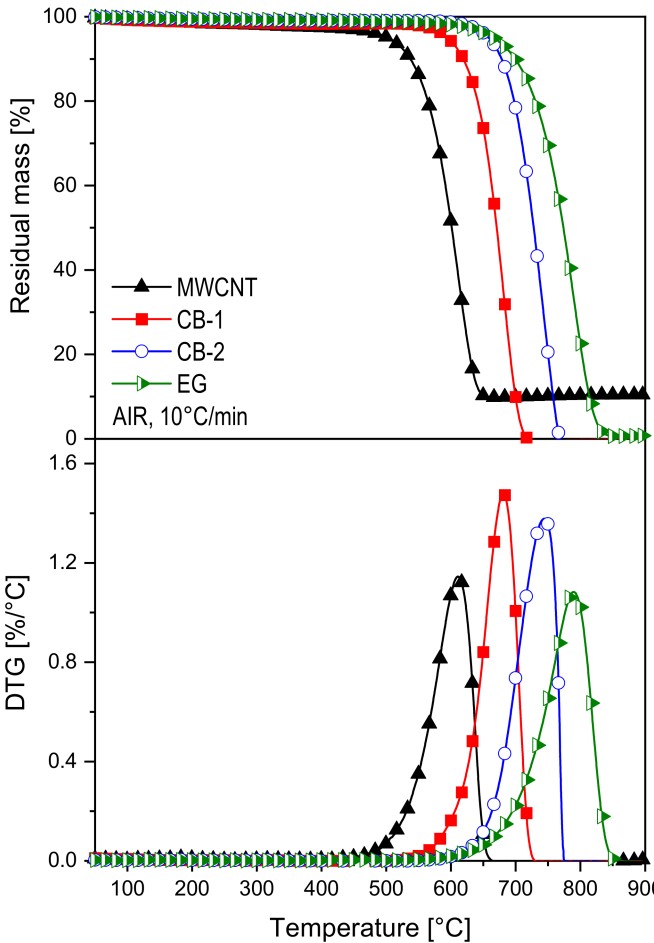

**Figure 1.** Thermogravimetric analysis of the carbon nanostructures in oxidative atmosphere.

DC electrical conductivity of the produced nanocomposites as a function of additives and weight content was evaluated and the results are reported in Figure 2. The most relevant outcome is related to the electrical percolation threshold, which varies with the selected carbon nanostructures.

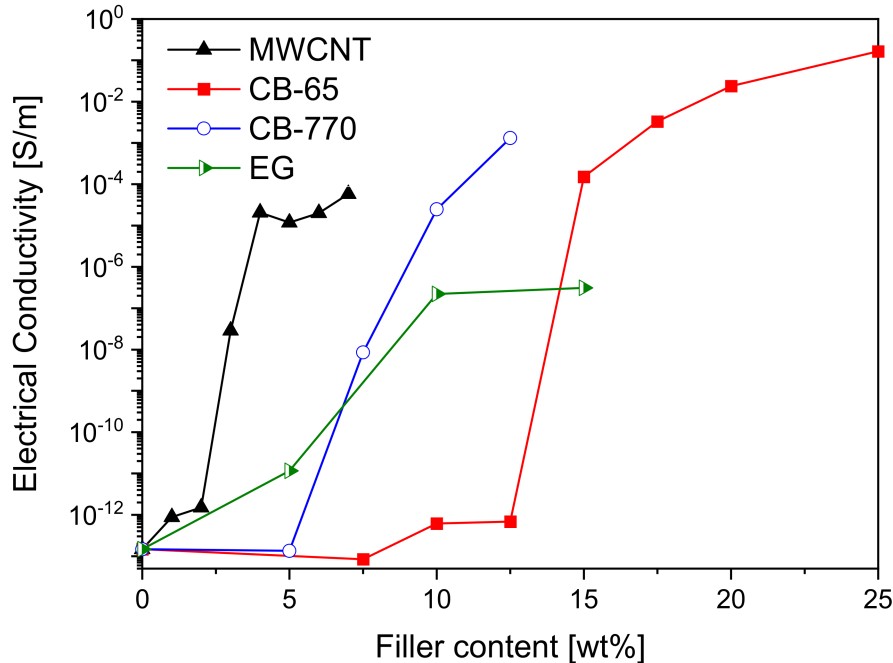

**Figure 2.** DC electrical conductivity of carbon-based nanocomposites at different additives content.

MWCNT nanocomposites pass from an electrical insulating behavior to a conductive one with a MWCNT content, which ranges between 2 wt% and 4 wt%. Globally, MWCNT-based nanocomposites reach an electrical conductivity plateau at $10^{-7}$–$10^{-6}$ S/m. CB-770 and EG nanocomposites turn from an insulating to a conductive electrical behavior with an additive content in the range 5–10 wt%, CB-770 nanocomposites reach $10^{-5}$ S/m as a maximum conductivity. Eventually, CB-65 nanocomposites show the highest percolation threshold value (in the range of 12.5–15 wt%) and the highest electrical conductivity ($10^{-3}$ S/m) with the highest used additive content (25 wt%).

The obtained results are in agreement to outcomes reported in the scientific literature. Studies demonstrated that the use of nanostructures with an aspect ratio greater than 1, like MWCNTs or EG nano-platelets, drastically reduces the percolation threshold [37–41]: higher surface areas favor the creation of nanostructures contacts, which allow electron conduction and tunneling between particles [23,42]. On the other hand, with spherical nanoparticles like CB, the construction of a continuous conductive path is more difficult and, for this reason, percolation thresholds are reached at higher additive contents if compared with MWCNTs or EG [43]. Nevertheless, CB with a larger structure, like CB-770, shows excellent conductive properties [44].

Figure 3 reports the AC conductivity of the studied nanocomposites as a function of frequency, in the range 10 Hz–1 MHz. If compared with the dielectric behavior of neat PP, the AC conductivity values of all the nanocomposites increases with the increase of the nanoadditive content. The AC conductivity of MWCNT nanocomposites, reported in Figure 3a, shows a plateau in the low frequency region beyond a content of 5 wt%, until a critical frequency is reached. The value of this critical frequency increases with increasing the MWCNT content. Beyond this frequency, the conductivity starts to increase linearly in a log–log scale. Similar results have been obtained with CB-65 and CB-770 nanocomposites. These results confirm that a conductive nature is leading at low frequency, whereas at higher frequency a capacitive nature occurs.

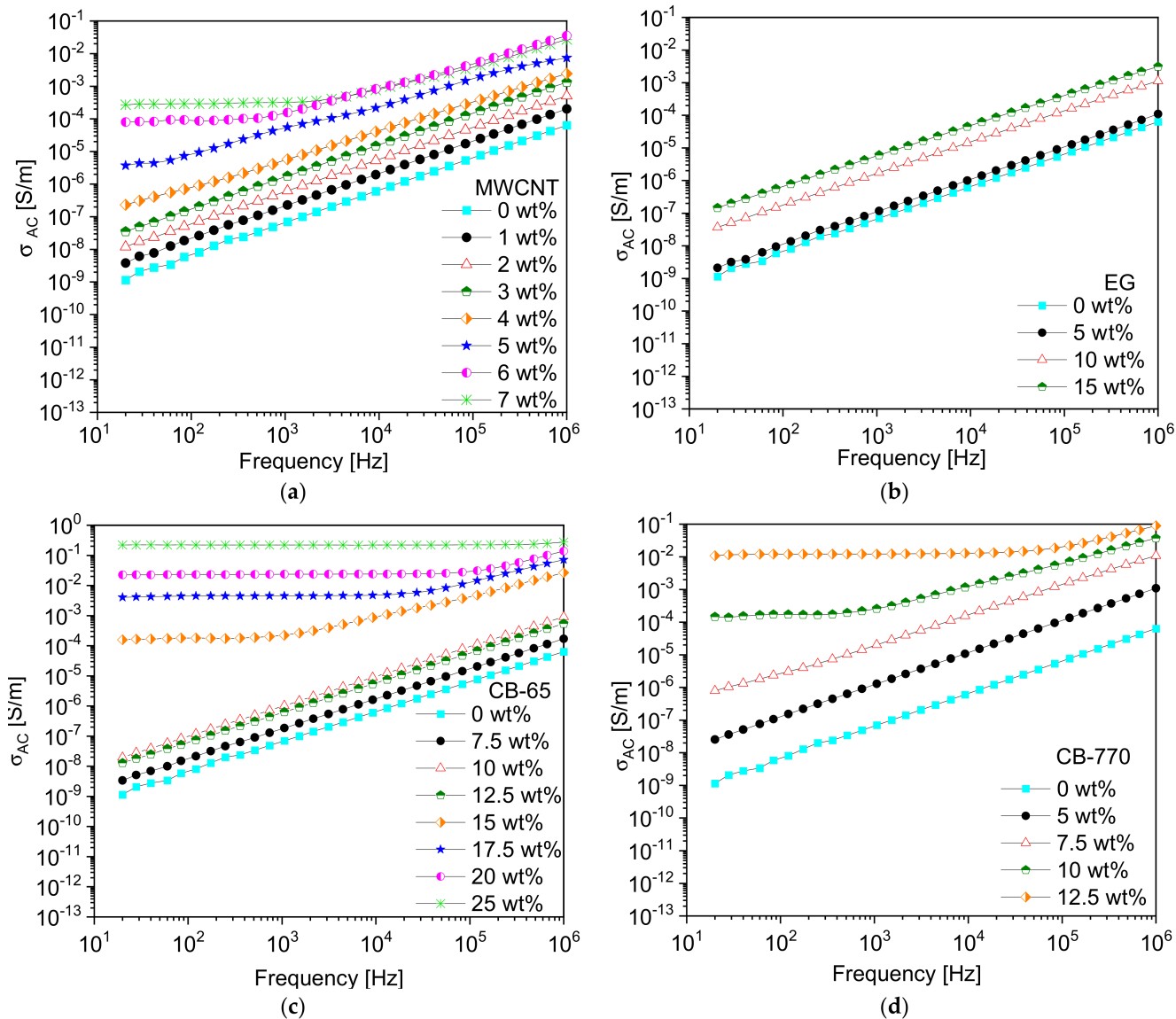

**Figure 3.** AC electrical conductivity σ_AC of (**a**) MWCNT, (**b**) EG, (**c**) CB-65, (**d**) CB-770 nanocomposites, respectively.

The described behavior is representative of the samples over the percolation threshold, and it can be explained by the current flow nearly totally through the nanoadditives network, which acts as a resistive path. On the other hand, when the frequency grows, the more capacitive parts, both the polymer matrix and the nanoparticles/polymer/nanoparticles contact points, give their own contribution in the increase of the global conductance of the system. The overall behavior of the σ_AC can be described as the superposition in the nanocomposites of both the capacitive and resistive components [33,34]. Conversely, no low frequency plateau is present in the EG formulations. In this case, the ohmic component of the conductivity is too low to be relevant in the global frequency range.

Figure 4 shows the results of the AC dielectric test in the frequency range of 1 MHz— 1 GHz, as the dependence of the real part of the dielectric constant (ε') of the nanocomposites with frequency, as a function of the used nanoadditives. The importance of studying ε' in this frequency range resides is the possibility of understanding the presence of polarization of the diverse dipoles induced by the different nanoadditives in the PP polymer matrix. These dipoles are typically due to the intrinsic polarity of molecular moieties, polarities related to the carbon nanoadditives and to the nanoadditives-polymer interphase [35].

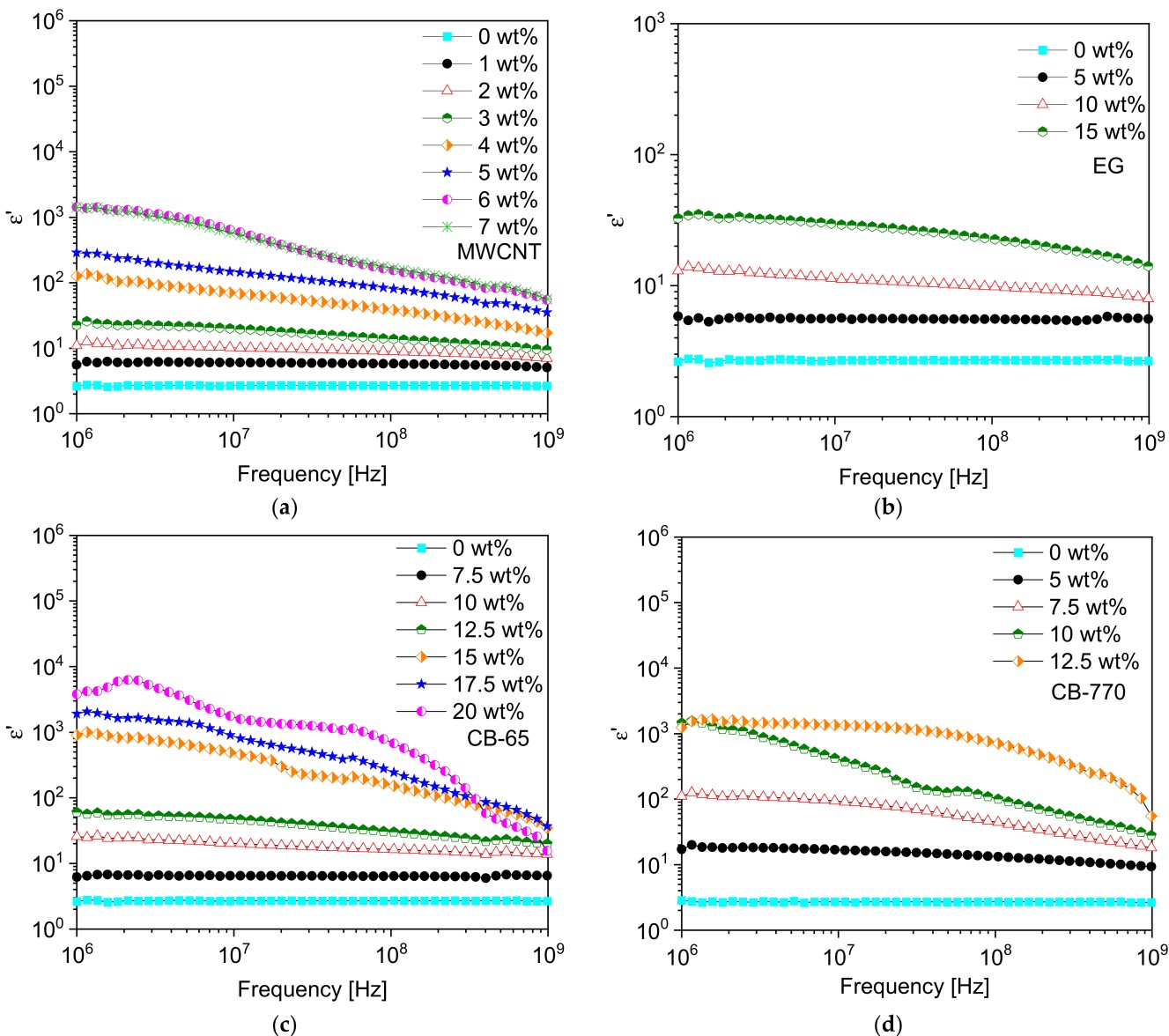

**Figure 4.** Real permittivity $\varepsilon'$ of (**a**) MWCNT, (**b**) EG, (**c**) CB-65, (**d**) CB-770 nanocomposites, respectively.

Neat PP has a low dielectric constant (2.2), which is independent by the frequency. For all the studied nanocomposites, the values of $\varepsilon'$ increase with increasing the carbon nanostructure content. Thus, the polarizability increases with increasing the nanoadditives content. This effect is stronger for MWCNT and CB-770 nanocomposites.

Furthermore, an increasingly higher dependence of dielectric constant with frequency is observed with increasing nanoadditives content. This could be attributed to the polarization effect of the conductive carbon-based nanostructures [45]. These results are consistent with what was obtained at low frequency in the dielectric spectroscopy and they are strictly related to the previous DC measurements.

As already cited, dissimilar electrical behavior of carbon-based nanocomposites can be ascribed to differences in aspect ratio and morphology of the several adopted nanoadditives. Figure 5 shows the SEM images at two different magnifications of MWCNT (a,b), EG (c,d), CB-65 (e,f), and CB-770 (g,h) nanocomposites, respectively.

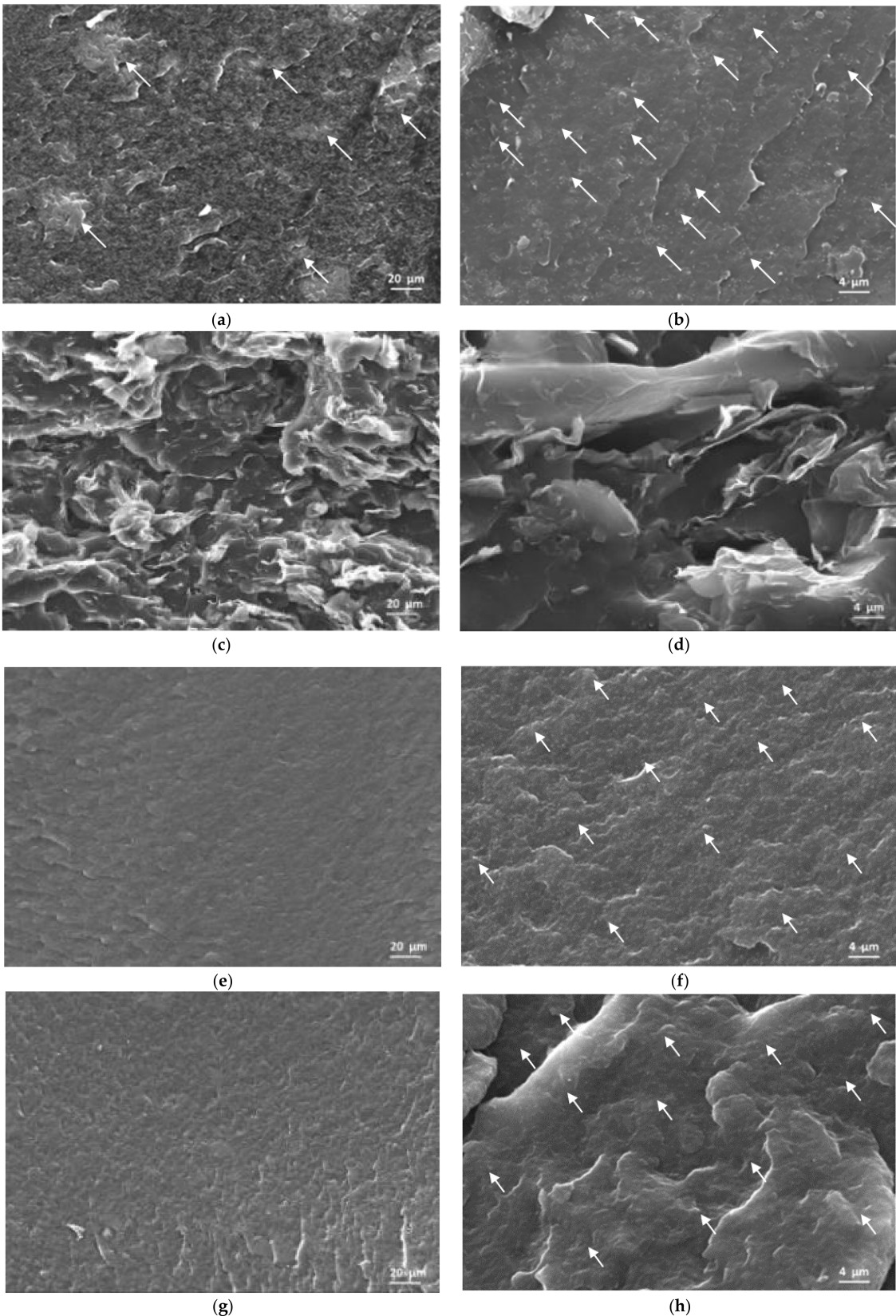

**Figure 5.** Scanning electron microscopy of (**a**) 5 wt% MWCNT (1.0× magnification), (**b**) 5 wt% MWCNT (5.0× magnification), (**c**) 15 wt% EG (1.0× magnification), (**d**) 15 wt% EG (5.0× magnification), (**e**) 17.5 wt% CB-65 (1.0× magnification), (**f**) 17.5 wt% CB-65 (5.0× magnification), (**g**) 10 wt% CB-770 (1.0× magnification), (**h**) 10 wt% CB-770 (5.0× magnification), espectively.

Taking into account MCWNT nanocomposites (Figure 5a,b), it can be observed that micrometric clustering formations are present (around 20–40 microns), but a well-dispersed submicronic population of agglomerates is also detectable (Figure 5a). Furthermore, analyzing the inner part of the clustering formation (Figure 5b), it can be noticed that MWCNTs appear completely immersed and dispersed in the polymeric matrix. Therefore, the clusters are not isolated entities in relation to the matrix, but the polymer reaches the internal areas of the carbon nanotubes clusters. This aspect can favor the creation of a percolative conductive network, in which the transfer of electrons occurs, increasing the electrical conductivity in the PP insulating matrix.

Considering EG nanocomposites (Figure 5c,d), a lamellar structure, in which all layers are stacked on top of each other, is detectable. This arrangement is typical for graphite-based additives and, based on the exfoliation level, a multi-layer traditional graphite can be obtained. Single lamellas appear oriented in the horizontal flux direction (Figure 5d).

Finally, for what concerns both CB nanocomposites, a well-dispersed morphology can be detected. No CB micro-agglomerates are visible on the analyzed surface. Moreover, any preferential orientation of the nanoadditive in the polymer matrix is observable. These statements can be considered valid for both CB-65 (Figure 5e,f) and CB-770 (Figure 5g,h) nanocomposites.

In order to have a complete overview of the carbon-based nanocomposites properties, their thermal conductivity was evaluated. Figure 6 displays the thermal conductivity as a function of the different nanoadditives content in the produced nanocomposites. Neat PP shows a thermal insulating behavior (thermal conductivity of 0.2 W/mK) and the presence of both MWCNT and CB (CB-65 and CB-770) in the final formulations does not significantly modify this value, even at the highest contents. A slight increase in thermal conductivity with the increase of the nanoadditives quantity is visible, but the obtained values remain in the thermal insulating range for all the samples as well.

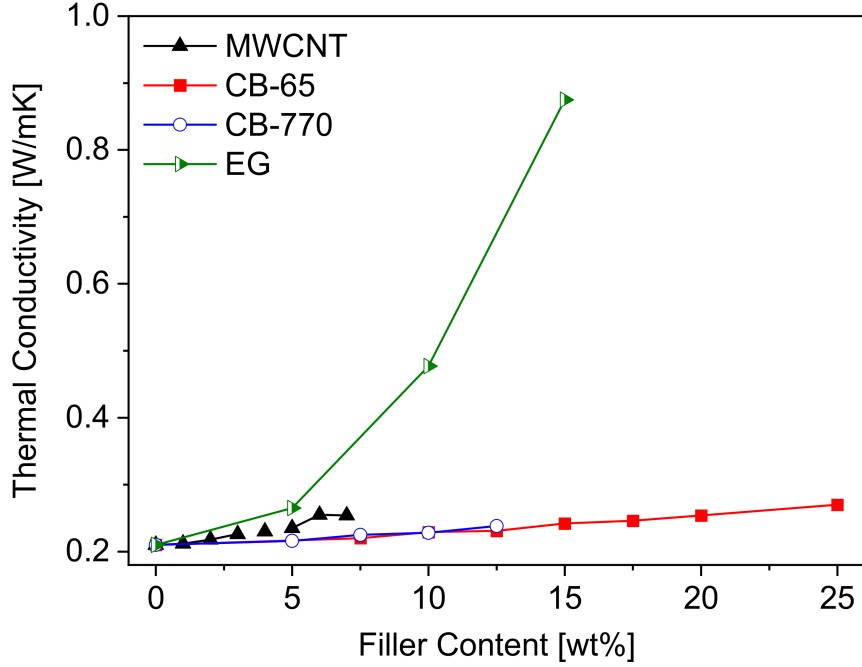

**Figure 6.** Thermal conductivity of carbon-based nanocomposites at different additives content.

Despite the potentialities for these nanoadditives in terms of thermal conductivity, the resulting enhancement of this parameter in the nanocomposites is not marked [46,47]. Zhong et al. [48] try to explain this phenomenon with the presence of a high number of poor coupling in the vibrational modes at the interfaces points between polymer and nanoadditives or nanoadditive and nanoadditive, due to the transfer method of the thermal

energy in the form of phonons. These weak connections reduce drastically the overall thermal conductivity of the system. Moreover, as reported in literature, MWCNT and CB contribute more to electrical conductivity rather than to thermal conductivity [3,14,48].

Conversely, if compared with all the other used nanoadditives, the addition of the EG leads to an increment in thermal conductivity, reaching values around 1 W/mK. Moreover, it should be noted that the measurements have been performed in the through-thickness direction (due to the used experimental setup), while, as demonstrated with SEM analysis previously discussed, EG particles are mostly oriented in the longitudinal direction, which can be related to a higher thermal conductive efficiency.

Finally, a mechanical characterization was performed. The produced nanocomposites have been tested according to tensile and notched Izod impact tests. Table 2 reports the obtained results.

**Table 2.** Mechanical results (tensile and Izod impact tests) of carbon-based nanocomposites at different additives content.

| Materials | Tensile Test | | | | | IZOD Notched Impact Test |
|---|---|---|---|---|---|---|
| | E (MPa) | $\sigma_y$ (MPa) | $\varepsilon_y$ (%) | $\sigma_b$ (MPa) | $\varepsilon_b$ (%) | Resilience (kJ/m$^2$) |
| Neat PP | $1230 \pm 30$ | $29.7 \pm 0.3$ | $9.5 \pm 0.1$ | $18 \pm 2$ | $500 \pm 100$ | $3.4 \pm 0.5$ |
| 1 wt% MWCNT | $1210 \pm 70$ | $29.9 \pm 0.2$ | $9.8 \pm 0.1$ | $18 \pm 1$ | $400 \pm 100$ | $3.5 \pm 0.4$ |
| 2 wt% MWCNT | $1220 \pm 80$ | $29.8 \pm 0.2$ | $9.9 \pm 0.1$ | $17 \pm 2$ | $400 \pm 80$ | $3.8 \pm 0.2$ |
| 3 wt% MWCNT | $1250 \pm 60$ | $30.3 \pm 0.4$ | $10.3 \pm 0.2$ | $16 \pm 2$ | $390 \pm 100$ | $3.9 \pm 0.4$ |
| 4 wt% MWCNT | $1367 \pm 24$ | $30.7 \pm 0.2$ | $10.2 \pm 0.1$ | $17 \pm 7$ | $25 \pm 6$ | $3.7 \pm 0.4$ |
| 10 wt% EG | $1590 \pm 40$ | $24.4 \pm 0.1$ | $6.8 \pm 0.1$ | $21.5 \pm 0.1$ | $13.0 \pm 0.5$ | $2.7 \pm 0.1$ |
| 15 wt% EG | $1860 \pm 60$ | $24.6 \pm 0.1$ | $5.6 \pm 0.1$ | $22.4 \pm 0.1$ | $9.5 \pm 0.2$ | $3.4 \pm 0.8$ |
| 15 wt% CB-65 | $1480 \pm 50$ | $30.3 \pm 0.2$ | $7.7 \pm 0.1$ | $29.2 \pm 0.3$ | $9.8 \pm 0.4$ | $2.4 \pm 0.3$ |
| 17.5 wt% CB-65 | $1510 \pm 50$ | $29.7 \pm 0.1$ | $6.5 \pm 0.1$ | $29.5 \pm 0.3$ | $6.6 \pm 0.2$ | $2.5 \pm 1.4$ |
| 20 wt% CB-65 | $1540 \pm 30$ | $28.9 \pm 0.6$ | $5.3 \pm 0.2$ | $28.7 \pm 0.5$ | $5.3 \pm 0.2$ | $1.6 \pm 0.1$ |
| 25 wt% CB-65 | $1610 \pm 60$ | $29.0 \pm 0.1$ | $4.9 \pm 0.1$ | $29.0 \pm 0.1$ | $4.9 \pm 0.1$ | $1.3 \pm 0.2$ |
| 5 wt% CB-770 | $1360 \pm 40$ | $30.5 \pm 0.3$ | $9.0 \pm 0.2$ | $14 \pm 5$ | $300 \pm 200$ | $2.3 \pm 0.4$ |
| 7.5 wt% CB-770 | $1440 \pm 24$ | $30.7 \pm 0.1$ | $7.7 \pm 0.1$ | $28 \pm 1$ | $11 \pm 1$ | $1.5 \pm 0.2$ |
| 10 wt% CB-770 | $1540 \pm 40$ | $30.8 \pm 0.1$ | $6.8 \pm 0.1$ | $29.9 \pm 0.4$ | $8.2 \pm 0.2$ | $1.4 \pm 0.2$ |

The presence of MWCNTs slightly affects the mechanical properties of the nanocomposites, in which they are embedded. More in detail, a slight increase of the elastic modulus and the tensile strength can be observed, with only a limited reduction of the elongation at break (significant only at 4 wt% content). The presence of MWCNTs also slightly enhances the impact strength, which is remarkable, since MWCNTs are not functionalized to make their interface with the polymer chains stronger [49–51].

On the other hand, the addition of EG, CB-65, and CB-770 leads to an increment in elastic modulus if compared with the neat PP and a reduction of the elongation at break, with no significant effect of the tensile strength. Furthermore, the addition of EG and both CB-65 and CB-770 leads to a reduction of the impact strength, with is increasingly lower at the higher amounts. It should be noted that these results are due to the higher amount of additive needed for overcoming the electrical (and thermal in case of EG) percolation threshold, which is supposed to be the main reason for using them in a polymer [52,53]. Moreover, as in the case of MWCNTs, no functionalization has been performed on these additives to create a stronger interface with the polymer matrix, resulting in a weak mechanical stress transfer between the matrix and the particles surface.

## 4. Conclusions

This study reports the results on the development of PP nanocomposites filled with different carbon-based nanoadditives, namely multi-walled carbon nanotubes (MWCNTs), expanded graphite (EG), carbon black at low surface area (CB-65), and carbon black at

higher surface area (CB-770). The added content is different for every additive, as it has been set in the range of the typical values of their electrical percolation threshold. The nanocomposite mixtures have been obtained by a melt extrusion process. The produced nanocomposites have been used to obtain the samples by injection molding. The morphological study performed by SEM has showed that in all cases an even dispersion has been obtained.

The DC electrical characterization showed that for all the nanoadditives the electrical percolation threshold is clearly visible. In particular, MWCNTs have the lowest value (in the range of 2–4 wt%), and the CB-65 has the highest. EG showed a lower conductivity value, beyond the percolation threshold, with respect to all the other nanoadditives.

The AC ($10^1$–$10^6$ Hz as frequency range) electrical conductivity has shown that there is a lower dependency with frequency with increasing the content of the nanoadditives in the case of MWCNT, CB-65, and CB-770, which is usually related to a shift from a fully capacitive to a resistive ohmic behavior. On the other hand, EG nanocomposites show a linear (log-log scale) dependency of the AC conductivity in the whole studied frequency range. This can be related to the lower DC conductivity beyond percolation, which seems not to allow the shift from a capacitive to an ohmic behavior.

The real part of the dielectric constant, which has been studied in the $10^6$–$10^9$ Hz frequency range, showed that in all cases there is an increase of the of the $\varepsilon'$ values, which is more pronounced for MWCNT, CB-65, and CB-770 nanocomposites.

The thermal conductivity of PP is significantly influenced only when EG is added, while no detectable results have been obtained with the other nanoadditives.

As for the mechanical properties, only MWCNTs allow the attainment of a slight increase of tensile and impact strength with no substantial reduction of the elongation at break. The other nanoadditives lead to a stronger increase of the elastic modulus, but with a significant reduction of elongation at break and impact strength. This is clearly due to the higher content need to overcome the percolation threshold in the case of EG, CB-65, and CB-770, with respect to MWCNTs.

**Author Contributions:** Conceptualization, M.M., M.Z., A.F. and I.A.; Methodology, M.M., M.Z., A.F. and I.A.; Validation, M.M. and M.Z.; Formal Analysis, M.M., A.F. and I.A.; Resources, M.Z.; Data Curation, M.Z.; Writing—Original Draft Preparation, M.M., M.Z. and I.A.; Writing—Review & Editing, M.M., M.Z., A.F. and I.A.; Supervision, M.M., A.F. and L.T.; Project Administration, M.Z. All authors have read and agreed to the published version of the manuscript.

**Funding:** This research received no external funding.

**Institutional Review Board Statement:** Not applicable.

**Informed Consent Statement:** Not applicable.

**Conflicts of Interest:** The authors declare no conflict of interest.

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
