# Peer review of "Effect of Filler Morphology on the Electrical and Thermal Conductivity of PP/Carbon-Based Nanocomposites"

_jcs, doi:10.3390/jcs5080196_

Round 1

Reviewer 1 Report

The manuscript entitled “Effect of filler morphology on the electrical and thermal conductivity of PP/carbon-based nanocomposites” investigated the electrical and thermal behavior of the PP/Carbon-fillers. Three nanofillers (CNTs, Graphene, CB) were used to produce the nanocomposites using a twin extruder. Overall, this study provides comprehensive research on the fundamental physical properties of PP composites. However, before publication, the following suggestions should be considered:

  1. In the abstract or the conclusion, the authors should highlight the breakthrough or the innovation of this study, or at least make a comparision with other literature discussing PP composite.

  1. Fig.5. The authors should highlight the fillers on the Figures since they are not noticeable now in the manuscript. Furthermore, the necessity of the Fig.5 a,c,e,f should be reconsidered.

  1. Fig. 2 and Fig.6, please show the error bar, or point out how many samples were measured to obtain the data points.

  1. Fig. 3 (b) The right axis is missing.

  1. Several related works of literature should be cited: Composites Science and Technology 150 (2017) 24-31, Applied Nanoscience (2018) 8:2071–2075.

Reviewer 2 Report

I found some symbolic errors in line 103 and 110. 

The sentences In line 236, "No micro-aggromerates...." is confusing, I suggest the rewiriting of this sentence.

Overall, the results are good. However, it need more supporting discussion for the concusion based on the results.

I found the morphology-based discussion using SEM images.

If the morphology-based discussion is exactly correct, one could get a different results of the different morphology sample using same materials. Is it sound ?

Reviewer 3 Report

This study reports a comparative study on the effect of different carbon 
nanostructures on the electrical, thermal conductivity and mechanical properties of the PP nanocomposites in which they are embedded. 

The manuscript  is well-written and most of the claims are firmly supported, therefore, I recommendation publication without revision. 
